# Mechanical Properties and Fatigue Life Estimation of Selective-Laser-Manufactured Ti6Al4V Alloys in a Comparison Between Annealing Treatment and Hot Isostatic Pressing

**DOI:** 10.3390/ma18153475

**Published:** 2025-07-24

**Authors:** Xiangxi Gao, Xubin Ye, Yuhuai He, Siqi Ma, Pengpeng Liu

**Affiliations:** 1Aero Engine Corporation of China Beijing Institute of Aeronautical Materials, Beijing 100095, China; yexubin2000@163.com (X.Y.); heyuhuai@163.com (Y.H.); 2Beijing Key Laboratory of Aeronautical Materials Testing and Evaluation, Beijing 100095, China; 17316088621@163.com; 3Key Laboratory of Aeronautical Materials Testing and Evaluation, Aero Engine Corporation of China, Beijing 100095, China; 15611789956@163.com

**Keywords:** selective laser melting, Ti6Al4V alloy, heat treatment, mechanical property, fatigue origin, fatigue life evaluation

## Abstract

Selective laser melting (SLM) offers a novel approach for manufacturing intricate structures, broadening the application of titanium alloy parts in the aerospace industry. After the build period, heat treatments of annealing (AT) and hot isostatic pressing (HIP) are often implemented, but a comparison of their mechanical performances based on the specimen orientation is still lacking. In this study, horizontally and vertically built Ti6Al4V SLM specimens that underwent the aforementioned treatments, together with their microstructural and defect characteristics, were, respectively, investigated using metallography and X-ray imaging. The mechanical properties and failure mechanism, via fracture analysis, were obtained. The critical factors influencing the mechanical properties and the correlation of the fatigue lives and failure origins were also estimated. The results demonstrate that the mechanical performances were determined by the *α*-phase morphology and defects, which included micropores and fewer large lack-of-fusion defects. Following the coarsening of the *α* phase, the strength decreased while the plasticity remained stable. With the discrepancy in the defect occurrence, anisotropy and scatter of the mechanical performances were introduced, which was significantly alleviated with HIP treatment. The fatigue failure origins were governed by defects and the *α* colony, which was composed of parallel *α* phases. Approximately linear relationships correlating fatigue lives with the X-parameter and maximum stress amplitude were, respectively, established in the AT and HIP states. The results provide an understanding of the technological significance of the evaluation of mechanical properties.

## 1. Introduction

Ti6Al4V alloy has been extensively designed for aerospace parts due to its characteristics of elevated specific strength, excellent corrosion resistance, and weldability [1,2,3]. However, its unfavorable performance, such as high melting point and deformation resistance, exerts adverse effects on traditional manufacturing and machining, especially for complex parts with cavity, thin-wall, or lattice structures. Currently, the advanced selective laser melting (SLM) technique, as one approach in additive manufacturing, has become a research target for fabricating metal parts in manufacturing industries [4,5].

SLM adopts a focused high-energy laser beam for selectively melting powder layer by layer, profitably fabricating dense titanium alloy parts with a complex geometry, high-dimensional accuracy, and better surface finish [6,7]. Prepared Ti4Al4V alloy in the ‘as-built’ state normally presents a non-equilibrium microstructure consisting of residual stress and acicular martensite, which is attributed to extremely rapid melting and cooling solidification [8,9,10]. Simultaneously, process-induced defects always survive from the specific metallurgical period [11,12,13,14]. Either of the above ingredients results in a reduction in mechanical properties, and these adverse effects can be alleviated by process optimization and some post-heat treatments [15,16,17]. Among them, annealing treatment (AT) and hot isostatic pressing (HIP) have been regarded as efficient routes for improving microstructural and mechanical properties from an unbalanced state. With an appropriate AT regime, the static performance achieves an excellent combination of a yield strength of ~980 MPa and an elongation of ~15% [15,18]. Additionally, HIP equally brings benefits, raising the fatigue strength up to ~700 MPa due to the elimination of intrinsic defects and the alteration of the initiation mechanism [19,20]. Previous research has mainly focused on exploiting the metallurgical quality and mechanical properties through altering the heat treatment modes, and a performance comparison of the AT and HIP conditions is still lacking. Furthermore, specimen orientation still has an effect on the thermal diffusion and morphology of the molten pool due to the various deflection angles of the laser beam, which often induces more defects in a larger melting layer. Thus, the property discrepancy and fatigue life estimation of the specimen orientations remain profound areas of investigation.

To compare the mechanical performances of SLM-manufactured Ti6Al4V alloys at different specimen orientations and AT and HIP conditions, the microstructures and defect characteristics were observed. The static and dynamic properties were also investigated, particularly for estimating the correlation of the fatigue life and failure origin.

## 2. Experimental Details

### 2.1. Materials

Ti6Al4V powder, prepared by the atomization method, provides a spherical morphology with particle sizes of 10.8–53.2 µm, as shown in Figure 1a. An EP-M300 SLM device (Yijia 3D Company, Hangzhou, China) was used for fabricating cuboids (5 mm × 5 mm × 5 mm), horizontally built (H) specimens, and vertically built (V) ones (Φ15 × 72 mm) within an inert argon-gas protection atmosphere (Figure 1b). The optimal process parameters were adopted (i.e., laser power of 350 W, scanning speed of 1200 mm/s, scanning spacing of 100 µm, and layer thickness of 60 µm), coupled with a chessboard scanning strategy. The ‘as-built’ specimens were removed from the substrate by wire cutting and then subjected to the AT and HIP processes. The AT regime was confirmed at 800 °C for 2 h, followed by furnace cooling, while HIP was conducted at 920 °C and 150 MPa for 3 h, and then furnace cooling was conducted [16,20]. The heat-treated H and V specimens were machined into standard round bar and smooth hourglass shapes for static and dynamic testing, respectively, as shown in Figure 1c,d.

### 2.2. Microscopic Analysis

The microstructural characteristics of cuboids in the AT and HIP states were observed. The cuboids were polished using a SAPHIR 550 facility (QATM Company, Dusseldorf, Germany) and then etched with an acid etchant (HF:HNO_3_:H_2_O = 3:5:92) for 10–15 s. The microstructure of the top and lateral planes was recorded using a GX_51 optical microscope (Leica Company, Frankfurt, Germany) and a NanoSEM 450 scanning electron microscope (FEI Company, Hillsboro, OR, USA). The crystal orientation was determined by capturing Kikuchi band patterns with 0.4 µm step-by-step scanning via an electron-backscatter-diffraction (EBSD) system (FEI Company, Hillsboro, OR, USA).

Microdefect analysis of the machined specimens was performed using an X-ray imaging technique. A Phoenix X-ray computed tomography (XCT) system (GE company, Boston, MA, USA) was used with the following parameters combination: voltage of 130 kV, current of 80 µA, exposure time of 500 ms, and rotation step of 0.36°. The specimen centroid approached the X-ray source, achieving a high resolution of 5 µm. The available data were reconstructed and statistically analyzed with Avizo software 9.0 (FEI Company, Hillsboro, OR, USA). To eliminate image noise and artifact interference, extracted microdefects that were less than 2 × 2 pixels were ignored.

### 2.3. Mechanical Testing

Axial tensile testing was performed for the machined specimens in the AT and HIP states according to ASTM E8/E8M standards. An INSTRON machine (Instron Corporation, Boston, MA, USA) was used with an extensometer (Epsilon 3560) of a strain rate of 0.005 min^−1^ at room temperature. Three specimens per orientation were tested for achieving the average strength/plasticity of the material.

Axial high-cycle fatigue (HCF) testing was also conducted using a QBG-50 fatigue system (Qian Bang Company, Changchun, China) according to ASTM E466 standard. A sinusoidal wave profile with a frequency of 125 Hz and a stress ratio of *R* = 0.1 was applied at room temperature. As the specimens failed or the cyclic number (N) reached 10^7^ cycles, the HCF testing was stopped artificially to obtain the *σ*_max_-lg*N*_f_ data. The fatigue limit of 10^7^ cycles was taken from the mean value of the unfractured specimens. Every fracture surface was observed by SEM.

## 3. Experimental Results and Discussion

### 3.1. Microscopic Characteristics

In the AT state, the specimen’s lateral plane entirely consisted of columnar grains that penetrated multiple molten layers in a nearly parallel build direction (Figure 2a). These columnar grains measured approximately 400–500 µm in length. The top plane presents cross-sectional cellular grains with a width of 100 µm, exhibiting a noticeable anisotropic feature (Figure 2b). This specific microstructure arises from the oriented temperature gradient during the build period that drives grain extension growth, while the gradual reduction in the gradient disparity, coupled with energy accumulation effects, leads to the re-nucleation and growth of novel columnar grains from adjacent boundaries [21]. Microscopically, the acicular *α* phase with a width of 2–3 µm fills the entire grain, and they interweave intimately with each other (Figure 2c). This indicates that the ‘as-built’ acicular martensite *α*’ transformation takes place in situ through the annealing process, and, subsequently, it is predominantly replaced by the acicular *α* [18,22,23]. During the build process, phase nucleation occurs at the grain boundaries and then forms *α* colonies after the transition period, composed of parallel *α* phases that sprout in specific directions and extend across the partial width of the columnar grains. Substantial *α* colonies possess the common characteristic of a random orientation, and form negligible, weak textures, i.e., a <11$\overset{—}{2}$4> or <11$\overset{—}{2}$2> fiber texture (Figure 2d) [24,25].

In the HIP state, the lateral and top planes retain a columnar and cellular grain morphology, respectively (Figure 3a,b). Under the condition of high pressure (150 MPa), the columnar grains experience partial deformation that reduces the aspect ratio and induces a certain degree of damage to the grain boundaries. The microstructure is not altered on the basis of a temperature below the *β*-transus compared with annealing. However, upon exposure to the elevated heat treatment temperature (920 °C), the grains consist entirely of lath *α*, with the widths increasing up to 3–5 μm, and their interfaces become more distinct due to the complete decomposition of the martensite *α*’ (Figure 3c) [26]. The crystal orientation’s characteristics are basically consistent with those under annealing, but the weak texture is altered, attributable to grain deformation, i.e., a <0001> fiber texture (Figure 3d).

In addition, the metallograph occasionally exhibits a smooth and regular micropore morphology in the AT state (Figure 4a). XCT imaging analysis revealed that the optimization process resulted in improved metallurgical quality, with a high density exceeding an average level of 99.984%, while micropores were preserved within the specimen volume, having a random distribution characteristic (Figure 4b). Statistical data further indicate that the size of the majority of the micropores in both the H and V specimens with densities of 99.986% and 99.995% was below 50 µm in a given volume, but this does not exclude the possibility of fewer large defects (Figure 4c). In contrast, the H specimens possessed a notably higher occurrence of defects probably due to an extensive melting layer region that caused pronounced fluctuations in the powder deposition, melting pool, or build procedure, thereby hindering metallurgical defect control [27]. Furthermore, the specimens in the HIP state displayed superior density without abnormal indications via XCT imaging, suggesting effective closure of internal defects in the alloy (Figure 4d).

### 3.2. Tensile Properties

The horizontally and vertically built SLM specimens in the AT and HIP states had a common microstructural characteristic in spite of the discrepancies in the *α*-phase sizes and defect occurrences that inevitably produced an effect on the mechanical properties.

Figure 5a shows the stress–strain curves of the SLM-manufactured specimens. In the AT state, the specimens presented outstanding tensile properties, including a tensile strength of approximately 1030 MPa, a yield strength of about 955 MPa, an elongation of 16.2%, and an area reduction of 47.2% (Figure 5b,c). These performances are attributed to the development of a homogeneous microstructure that comprised acicular *α*. However, the H specimens exhibited lower plasticity values compared with the V ones, exhibiting obvious plastic anisotropy. This specific phenomenon could be attributed to a higher occurrence of dispersed defects involved in the fracture period of the H specimens (Figure 5d,e), thereby strengthening the alloy’s anisotropy level but having negligible effects on the strength. The findings on the fracture surfaces are in alignment with the XCT imaging analysis.

In the HIP state, the specimens retained an excellent combination of a tensile strength of approximately 935 MPa, yield strength of about 850 MPa, elongation of 16.3%, and area reduction of 44.1%. Compared with annealing, the coarsening of the lath *α* led to a reduction in strength while preserving the plasticity. Moreover, the reality of the defect closure after the HIP process significantly alleviated the alloy’s plastic anisotropy.

The representative tensile fractures in the AT and HIP states exhibited distinct fibrous and radial zones. The fibrous zones in the H and V specimens, respectively, exhibit elliptical (Figure 5d,g) and circular morphologies (Figure 5e,h), while the ellipse’s long axis aligns parallel to the build direction. This alignment is attributed to the higher crack propagation rate along the length of the columnar grain boundaries involved in the fracture period [28]. Microscopically, the fibrous zone in the annealed specimens displayed fine and deep dimples smaller than 10 µm (Figure 5f). Moreover, the HIP state exhibited relatively coarse and shallow dimples (Figure 5i), indicating the coalescence of microvoids primarily at the *α*-phase interfaces, leading to a ductile fracture pattern in the alloy.

### 3.3. Fatigue Properties

Figure 6 shows the fatigue *σ*_max_-lg*N*_f_ data of the SLM-manufactured specimens per the orientation in the AT and HIP states. The fatigue life demonstrated an increasing trend with a decreasing maximum stress amplitude. In the AT state, the specimens presented notable fatigue anisotropy and scatter. For instance, the fatigue limits at 10^7^ cycles of the H and V specimens were 487.5 MPa and 562.5 MPa, respectively. Moreover, the fatigue lifespan at several stress amplitudes potentially exceeded one million cycles. The above characteristics have been certified as correlating with the type of defects serving as fatigue crack initiation sites, specifically stemming from discrepancies in size, location, shape, and other related factors [29,30,31]. In the HIP state, abnormal fatigue performances was extremely alleviated, establishing an approximate linear relationship between *σ*_max_ and lg*N*_f_ (linearity, *r*^2^ = 0.62). The fatigue limits presented similar values between the H and V specimens (687.5 MPa and 675 MPa), and the lifespan became more concentrated, which is attributed to alteration of the fatigue crack initiation sites as a result of defect closure [26].

Figure 7 shows the fatigue fracture characteristics of the SLM-manufactured specimens per the orientation in the AT and HIP states. The fracture surfaces can be categorized into three distinct regions delineated by dotted lines—fatigue initiation zone, crack propagation zone, and instantaneous fracture zone (Figure 7a,g). The initial two zones were predominantly perpendicular to the applied stress, while the latter one exhibited a smooth 45° orientation to the principal stress. This behavior is attributed to the transition from a three-dimensional stress state to a plane stress state during the crack propagation period [32]. Through observation of the fatigue crack initiation sites, two different failure mechanisms were identified, surface and interior crack initiations, with the former commonly linked to a shorter fatigue life [33].

In the AT state, the dominant surface crack initiation accounted for 86% of the fractured specimens, consisting of two representative defect types—micropores and lack of fusion (LOF). Micropores may be caused by hollow powder or trapped gas within the molten pool (Figure 7b) [34]. The type of LOF consisting of unmelted powder particles results from an inadequate laser-energy-density input, which forms a layered structure perpendicular to the build direction, where its intricate morphology with elevated stress contributed to an inferior fatigue life (Figure 7d) [35]. Interior LOF could contribute to inducing crack initiation due to a sudden rise in stress concentration (Figure 7e). Notably, the SLM-manufactured specimens exhibited a higher tolerance for interior defects, as they had a slower crack propagation rate compared with surface defects [36]. Here, the fatigue initiation life, composed of the formation and propagation of small cracks, can be neglected because of the presence of these defects. In the HIP state, the surface crack initiation decreased to 65% of the fractured specimens, and *α* colonies forming oriented facets were found to be responsible for the failure origin instead of the defects (Figure 7h,j,k) [37]. These *α* colonies determine that the applied stresses are linearly linked to different fatigue lives, wherein the total fatigue life relates to the crack initiation and stable propagation stages.

The representative fatigue fractures in the AT and HIP states exhibited similarity in crack propagation and instantaneous fracture zones. In the crack propagation zone, slip bands formed at the crack tip due to the cyclic plastic strain, resembling continuous fatigue striations. This facilitates the penetration of a crack to the *α* phase, resulting in a transgranular fracture (Figure 7c,i). In the instantaneous fracture zone, the surface was characterized by numerous shallow dimples, indicating a ductile fracture with microvoid coalescence (Figure 7f,l).

### 3.4. Defect-Based Fatigue Life Evaluation

Precisely assessing the fatigue life of additive-manufactured metal alloys remains a significant challenge owing to various intricate factors. In the present work, a fractographic analysis revealed that three distinct crack initiation sites—micropore, LOF, and *α* colony—are intimately related to the fatigue life of SLM-manufactured specimens, wherein the *α* colony in the HIP state determines an approximate linear relationship between *σ*_max_ and lg*N*_f_. In order to assess the effect of the underlying geometric parameters of the defects on the fatigue life in the AT state, a novel X-parameter fatigue life prediction model was proposed, which has been applied to evaluate the fatigue life of electron-beam-manufactured Ti6Al4V welded joints [38]. In general, the X-parameter model can be written as follows:(1)$$X=\sigma_{\max}\cdot {\left(\sqrt{area}\right)}^{1/6}\cdot D^{\beta }/C^{\alpha }$$(2)D=(d−h)/d
where *σ*_max_ corresponds to the maximum stress amplitude. The Murakami parameter [39], √*area*, and the size of the nearest distance from the crack initiation contour to the specimen surface, *h*, determine the size and location of failure origins. To compare the values of √*area* and *h*, the surface and interior crack initiations can be simply identified, as show in Figure 8. *C* is the shape parameter that represents the circularity of the defect contour, which can be obtained by the software ImageJ. The more regular the shape, the greater the value of *C*. *D* is a location parameter that can be calculated from the specimen’s diameter, *d* and *h*. *α* and *β* are the optimal fitting parameters according to the X-lg*N*_f_ data, obtained using a specific Python program.

According to a fractographic analysis of the fractured specimens in the AT state, the experimental results and geometric parameters (√*area*, *C* and *D*) of the defects serving as the failure origins are summarized in Table 1. Furthermore, Figure 9 shows the calculated fatigue X-lg*N*_f_ data and fitting results of the SLM-manufactured specimens per the orientation. Compared with the *σ*_max_-lg*N*_f_ data, the fatigue scatter was significantly reduced (*r*^2^ = 0.92), suggesting the influence of the geometric parameters of the defects on the fatigue life. However, the X-parameter model was not suitable for fatigue life prediction that depended on the *α* colony in the HIP state. This complexity is, indeed, beyond the scope of the current study. Future work will address this important topic [40,41].

Despite the improved linearity in this study, it remains influenced by the following aspects: (i) extraction accuracy of the feature parameters from the defects; (ii) neglect of the influence of the three-dimensional morphology of the defects; (iii) neglect of the effect of persistent slip bands on the fatigue life evaluation, commonly associated with surface crack initiation; and (iv) disregard of other defects along the crack propagation route on the fatigue life.

## 4. Conclusions

In this study, the horizontally and vertically built SLM Ti6Al4V specimens subjected to heat treatments of annealing and hot isostatic pressing, including their microstructural and defect characteristics, were investigated. The strength/plasticity values, high-cycle fatigue properties, and failure mechanisms were obtained. The critical factors influencing the mechanical properties were identified, and the correlation between fatigue lives and failure origins was also estimated. The conclusions are as follows:The SLM-manufactured specimens presented similarities in their microstructures, consisting of columnar grains, an *α* phase, and weak textures in the AT and HIP states. In contrast, *α* phase coarsening and defect closures occurred in cases of elevated temperature and high pressure. Predominantly, dispersed micropores and fewer large LOFs resided in specimens in the AT state, and a higher occurrence of defects in the H specimens compared with the V ones was found.The tensile properties and failure mechanism were intricately correlated with the *α* phase and defects. The formation of an acicular *α* phase resulted in superior strength/plasticity, as well as a ductile fracture characterized by microvoid coalescence. The coarsening of the lath *α* decreased the strength but preserves the plasticity. The plastic anisotropy was significantly alleviated through the reality of defect closure.The fatigue life of the SLM-manufactured specimens in the AT state featured notable anisotropy and scatter but was significantly improved with the HIP treatment. The predominant surface crack initiation and interior crack initiation were identified in the HCF regime. Furthermore, the failure origins were governed by process-induced micropores, LOF, and an *α* colony composed of parallel *α* phases.Approximately linear relationships, X-lg*N*_f_ and *σ*_max_-lg*N*_f_, were, respectively, established in the AT and HIP states. An X-parameter fatigue life prediction model was proposed by considering the size, location, and shape of defects serving as failure origins. This approach effectively alleviates fatigue anisotropy and scatter.

## Figures and Tables

**Figure 1 materials-18-03475-f001:**
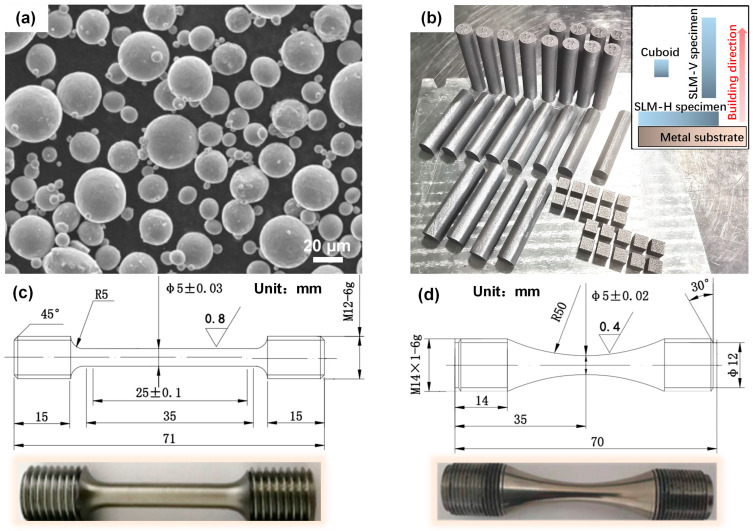
Schematic diagrams showing the specimen preparation: (**a**) Ti6Al4V spherical powder; (**b**) SLM-manufactured cuboids and H and V specimens; (**c**) round tensile specimen; (**d**) smooth hourglass fatigue specimen.

**Figure 2 materials-18-03475-f002:**
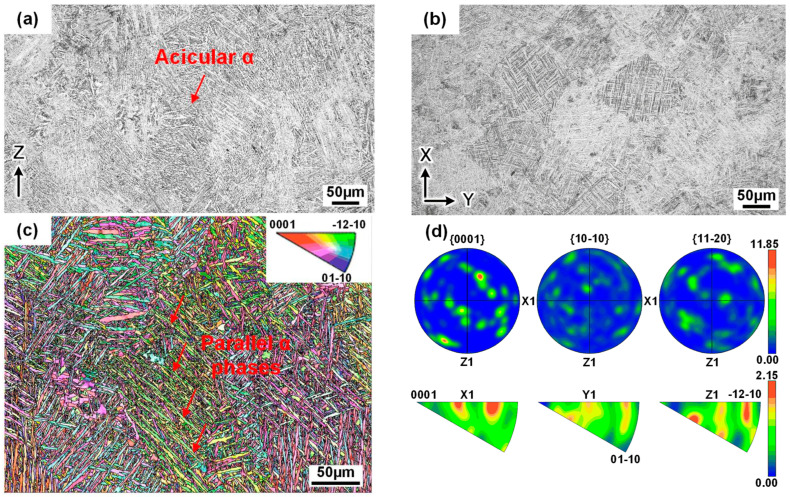
Microstructural characteristics of the SLM-annealed cuboid: (**a**) lateral plane; (**b**) top plane; (**c**) orientation map of the Ti-*α* phase; (**d**) corresponding pole and inverse pole figures.

**Figure 3 materials-18-03475-f003:**
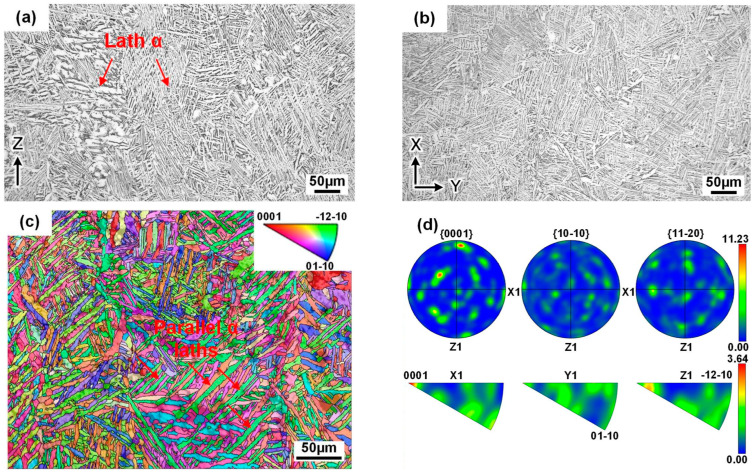
Microstructural characteristics of the SLM-HIPed cuboid: (**a**) lateral plane; (**b**) top plane; (**c**) orientation map of the Ti-*α* phase; (**d**) corresponding pole and inverse pole figures.

**Figure 4 materials-18-03475-f004:**
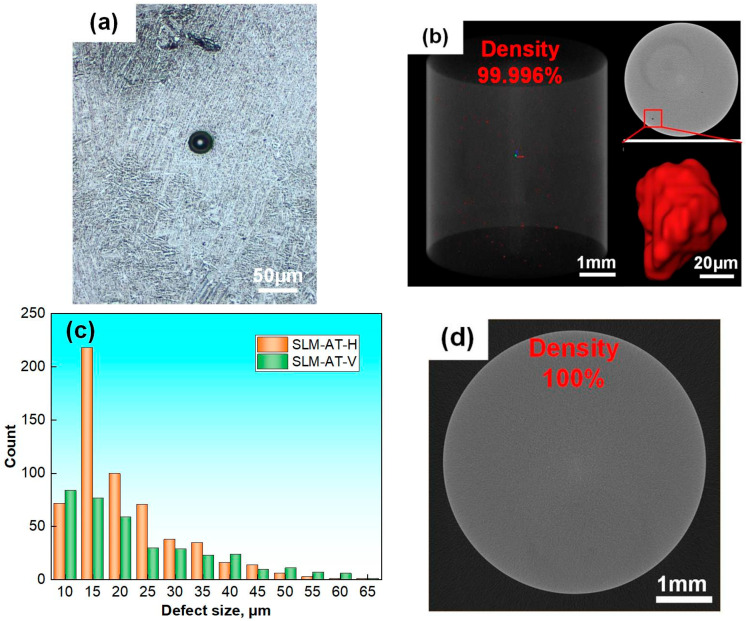
Microdefect characteristics of the SLM-annealed and -HIPed specimens: (**a**) metallograph of a micropore; (**b**) XCT imaging of the three-dimensional defect distribution and morphology; (**c**) statistical analysis of the defect sizes in the annealed H and V specimens; (**d**) typical XCT slice of the HIPed specimens.

**Figure 5 materials-18-03475-f005:**
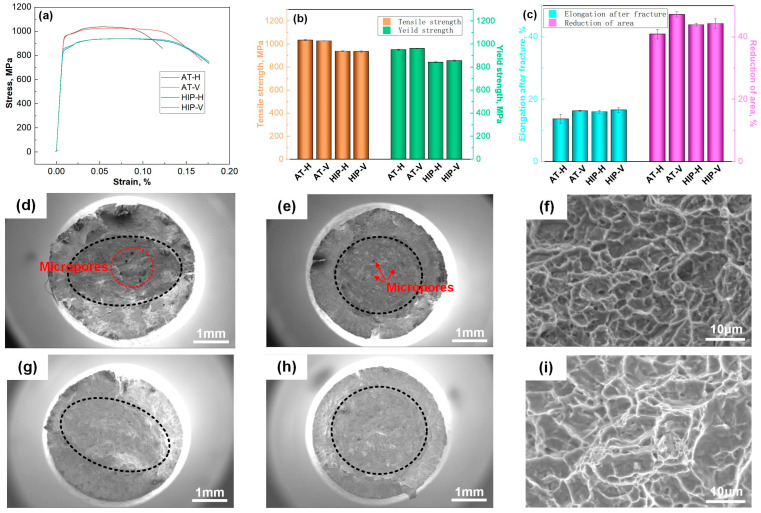
Tensile properties and fracture images of the SLM-annealed and -HIPed specimens at various specimen orientations: (**a**) stress–strain curves; (**b**) tensile and yield strengths; (**c**) elongation after fracture and reduction in area; (**d**) fibrous zone of an annealed H specimens; (**e**) fibrous zone of an annealed V specimen; (**f**) magnified dimples in the fibrous zone of an annealed specimen; (**g**) fibrous zone of an HIPed H specimen; (**h**) fibrous zone of an HIPed V specimen; (**i**) magnified dimples in the fibrous zone of an HIPed specimen. The black dotted lines represent the boundary between fibrous and radial zones.

**Figure 6 materials-18-03475-f006:**
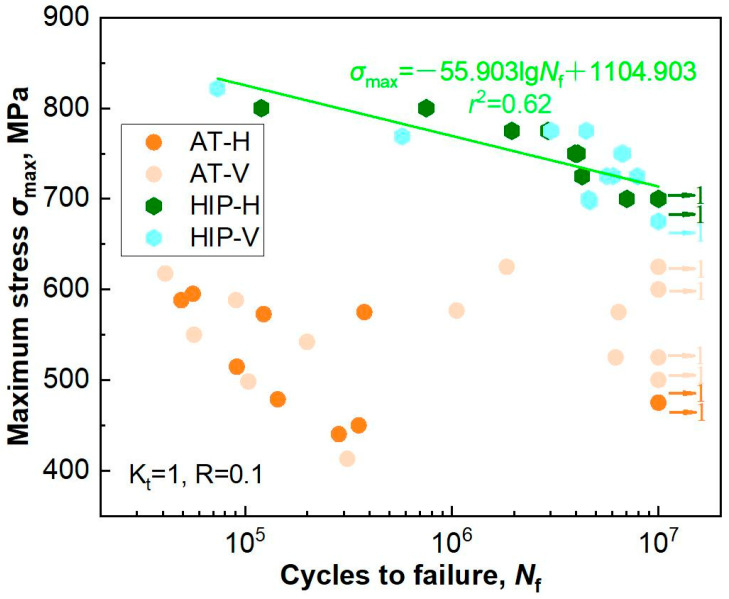
Fatigue properties of the SLM-annealed and -HIPed specimens at various specimen orientations. The arrows and the corresponding numbers denote non-fracture specimens with lives exceeding 10^7^ cycles.

**Figure 7 materials-18-03475-f007:**
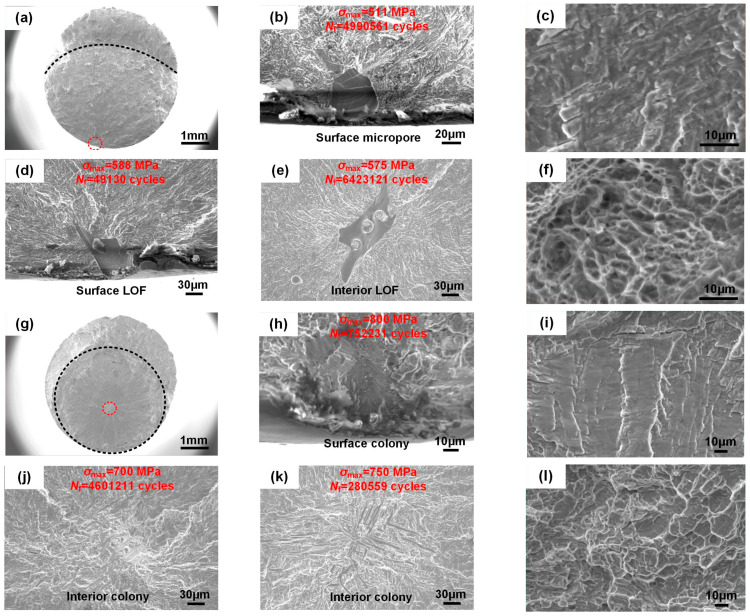
Fatigue fracture images and crack initiation sites of the SLM-annealed and -HIPed specimens at various specimen orientations: (**a**) fracture surface of an annealed specimen; (**b**) surface micropore; (**c**) magnification of the crack propagation zone of an annealed specimen; (**d**) surface LOF; (**e**) interior LOF; (**f**) magnification of the instantaneous fracture zone of an annealed specimen; (**g**) fracture surface of an HIPed specimen; (**h**) surface colony facet; (**i**) magnification of the crack propagation zone of an HIPed specimen; (**j**) interior colony facet; (**k**) interior colony facet; (**l**) magnification of the instantaneous fracture zone of an HIPed specimen.

**Figure 8 materials-18-03475-f008:**
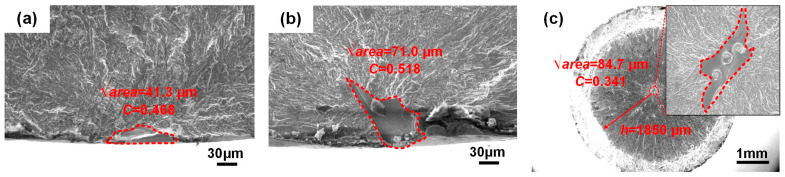
Geometric parameters of the defects in the SLM-manufactured specimens in the AT state: (**a**) surface micropore; (**b**) surface LOF; (**c**) interior LOF.

**Figure 9 materials-18-03475-f009:**
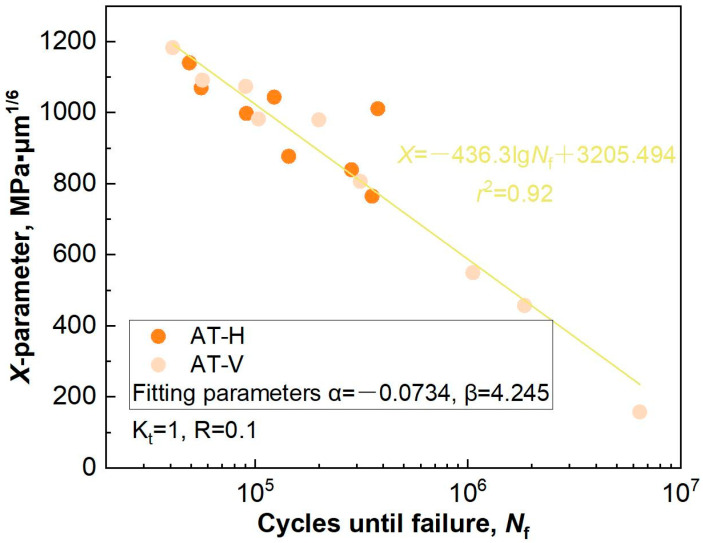
Fatigue life fitting results of the SLM-manufactured specimens in the AT state based on the X-parameter model.

**Table 1 materials-18-03475-t001:** Experimental results and geometric parameters of the defects.

Specimens	Maximum Stress Amplitude, *σ*_max_/MPa	Fatigue Life, *N*_f_/Cycles	Size, √*area*/µm	Location, *D*/µm	Shape, *C*
AT-H1	575.0	377,572	41.3	1	0.468
AT-H2	450.0	354,754	31.2	1	0.558
AT-H3	514.7	91,159	65.3	1	0.626
AT-H4	440.3	284,041	54.8	1	0.738
AT-H5	588.2	49,130	71.0	1	0.518
AT-H6	595.2	55,858	38.6	1	0.739
AT-H7	572.7	123,060	50.2	1	0.490
AT-H8	478.7	143,928	44.2	1	0.703
AT-V1	498.4	103,769	60.4	1	0.938
AT-V2	617.6	41,072	69.8	1	0.456
AT-V3	588.2	90,355	42.7	1	0.726
AT-V4	413.0	312,444	72.8	1	0.540
AT-V5	542.1	199,436	40.4	1	0.715
AT-V6	575.0	6,423,121	84.7	0.631	0.341
AT-V7	550.0	56,607	134.5	1	0.167
AT-V8	625.0	1,846,441	38.4	0.816	0.459
AT-V9	576.7	1,054,956	142.4	0.836	0.211

## Data Availability

The original contributions presented in this study are included in the article. Further inquiries can be directed toward the corresponding author.

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
