# Peer review of "Mechanical Properties and Fatigue Life Estimation of Selective-Laser-Manufactured Ti6Al4V Alloys in a Comparison Between Annealing Treatment and Hot Isostatic Pressing"

_materials, 2025, doi:10.3390/ma18153475_

Round 1

Reviewer 1 Report

Comments and Suggestions for Authors

In materials-3725856, Mechanical properties and fatigue life estimation of selective laser manufactured Ti6Al4V alloys in comparison of annealing treatment and hot isostatic pressing are described. This subject is actual and interesting, and actual measuring devices, methodology and (in part) references are used. The main goal of this research is analysis of SLM process parameters influenced on material structured, mechanical and fatigue properties porosity and so on. This topic is original and relevant, and completing of AM processing and Ti alloy features is specific gap in this area, because only little references are known.

The authors identified interesting structural and material relationships in comparison to known relationships in AD manufacturing. The conclusions are consistent with studies made, the arguments presented and questions discussed in this study. The tables and figs are interesting and informative.

However, some small remarks should be done:

1. The results of the statistical analysis of the research findings should be described in text and Figs.
2. The vast majority of references are 5-10 years old, so newer sources should be added.

Comments on the Quality of English Language

I can not be qualified expert for English language

Author Response

Comment 1: The results of the statistical analysis of the research findings should be described in text and Figs.

Response 1: Thanks for your valuable comments. The authors did indeed ignore the issue and did not explain this clearly. We have modified all figures and the corresponding figure legends in the paper, and enriched the analyzed results in text and figures, i.e., α phase size in section 3.1, statistical density in Fig. 4, and fitting formula in Fig. 6 and Fig. 9.

Comment 2: The vast majority of references are 5-10 years old, so newer sources should be added.

Response 2: Thanks for your suggestion. Revised accordingly. We have added the more recent references.

Reviewer 2 Report

Comments and Suggestions for Authors

The manuscript explores the mechanical properties and fatigue life of selective laser melted Ti6Al4V alloys under annealing and hot isostatic pressing treatments, with a focus on specimen orientation. This is a valuable contribution to the field of additive manufacturing. However, several areas require attention to enhance the manuscript’s rigor and clarity.

Comment 1: The qualitative descriptions of microstructure and defects are insightful, but the manuscript would be significantly strengthened by including more quantitative data. For example, in the microstructure analysis, details such as the average α lath thickness or grain size distribution could provide a clearer picture of the differences between AT and HIP conditions. Similarly, in the defect analysis, reporting porosity levels or average defect sizes would allow readers to better assess the impact of these features on mechanical properties.

Comment 2: The fatigue testing appears to be based on a limited number of specimens, which raises concerns about the robustness of the S-N curves, especially for a material known for scatter in fatigue performance. Expanding the dataset or discussing the limitations of the current sample size would improve the credibility of your fatigue life conclusions.

Comment 3: The X-parameter model for fatigue life prediction is a compelling addition, particularly for the AT condition where defects dominate crack initiation. However, its applicability to the HIP condition is questionable since crack initiation shifts to microstructural features (α colonies) rather than defects. A discussion on how the model performs in the HIP case or whether an alternative approach is needed would enhance this section.

Comment 4: The manuscript’s readability is hampered by grammatical errors and awkward phrasing. For instance, the title’s phrase "in comparison of annealing treatment and hot isostatic pressing" should be revised to "in comparison to annealing treatment and hot isostatic pressing." A thorough language review is necessary to ensure clarity and professionalism.

Comment 5: The introduction provides a solid background but could be improved by briefly explaining why build orientation affects SLM part properties. A sentence or two about how the layer-by-layer process influences defect distribution or anisotropy would help readers unfamiliar with additive manufacturing understand the study’s motivation.

Comment 6: In section 2.1, the experimental details mention cuboids and specimens, but their dimensions are not specified. 

Comment 7: The tensile properties section (3.2) reports strength and ductility values, but it’s unclear how many tests were conducted per condition. 

Comment 8: In the fatigue properties section (3.3), the method for determining fatigue limits (e.g., staircase method or curve fitting) is not specified. Clarifying this would aid in interpreting the reported values of 487.5 MPa and 562.5 MPa for AT specimens, and 687.5 MPa and 675 MPa for HIP specimens.

Comment 9: The X-parameter model in section 3.4 is well-presented, but the manuscript does not explain how the fitting parameters α and β were determined. Additionally, discussing potential sources of error in the model (beyond the two aspects already noted) would provide a more balanced evaluation of its effectiveness.

Comment 10: The references are relevant, but incorporating more recent studies on SLM Ti6Al4V fatigue behavior (e.g., from the last 2–3 years) could further contextualize your findings within the current state of the field.

Author Response

Comments 1: The qualitative descriptions of microstructure and defects are insightful, but the manuscript would be significantly strengthened by including more quantitative data. For example, in the microstructure analysis, details such as the average α lath thickness or grain size distribution could provide a clearer picture of the differences between AT and HIP conditions. Similarly, in the defect analysis, reporting porosity levels or average defect sizes would allow readers to better assess the impact of these features on mechanical properties.

Response 1: Thanks for your professional comments. We have modified the grain size and  acicular/lath α size in the microstructure analysis, and provided the density of H and V specimens in terms of X-ray imaging analysis in Section 3.1.

Comments 2: The fatigue testing appears to be based on a limited number of specimens, which raises concerns about the robustness of the S-N curves, especially for a material known for scatter in fatigue performance. Expanding the dataset or discussing the limitations of the current sample size would improve the credibility of your fatigue life conclusions.

Response 2: Thanks for your valuable suggestion. Experimental studies [1,2] have previously reported that the HCF life data of SLM manufactured Ti6Al4V alloy presented extensive scatter. In this study, we have raised the fatigue life data of limited samples at the stress ratio R of 0.1 in the AT and HIP status. Indeed, the testing data has a poor fitting effect due to few sample numbers in the AT status, thus we have deleted plotted fatigue curve in Fig. 6. However, we have still found the pronounced fatigue scatter of the testing data, and even the fatigue limit anisotropy of SLM-AT-H and SLM-AT-V. The fractured samples are capable of tomographic imaging of defects in advance, and these limited samples still can establish the relation of fatigue life and defect characteristic regardless of process and sample orientation.

[1] Yadollahi A, Shamsaei N. Additive manufacturing of fatigue resistant materials: Challenges and opportunities. Int J Fatigue 2017, 98: 14–31.

[2] Chen BQ, Wu ZK, Yan TQ, He Z, Sun BB, Guo GP, Wu SC. Experimental study on mechanical properties of laser powder bed fused Ti-6Al-4V alloy under post-heat treatment. Eng Fract Mech 2022, 261: 108264.

Comments 3: The X-parameter model for fatigue life prediction is a compelling addition, particularly for the AT condition where defects dominate crack initiation. However, its applicability to the HIP condition is questionable since crack initiation shifts to microstructural features (α colonies) rather than defects. A discussion on how the model performs in the HIP case or whether an alternative approach is needed would enhance this section.

Response 3: Thanks for your comments. The authors did indeed ignore the issue and did not explain this clearly. Due to the defect closure after the HIP treatment, the fatigue crack initiation sites alter from defects to α colonies. This phenomenon makes fatigue lifespan more concentrative, forming an approximate linear relationship between the applied maximum stress (σmax) and Logarithm Nf (lgNf). However, the total fatigue life in terms of α colonies comprises of the crack initiation and stable propagation stages, which is not appropriate for the X-parameter model for fatigue life prediction merely depending on the stable propagation stage due to the occurrence of defects. Therefore, we have added the sentence in Section 3.4 “However, the X-parameter model is not suitable for fatigue life prediction depending on the α colony in the HIP status. This complexity is indeed beyond the scope of the current study. Future work will address this important topic.”

Comments 4: The manuscript’s readability is hampered by grammatical errors and awkward phrasing. For instance, the title’s phrase "in comparison of annealing treatment and hot isostatic pressing" should be revised to "in comparison to annealing treatment and hot isostatic pressing." A thorough language review is necessary to ensure clarity and professionalism.

Response 4: We are very sorry for our poor English writing. We have pursued a native English scholar carefully checked the paper, correct grammar and rephrase with text that make it more accurate and concise. The modified version has been significantly improved from the perspective of logicality and readability.

Comments 5: The introduction provides a solid background but could be improved by briefly explaining why build orientation affects SLM part properties. A sentence or two about how the layer-by-layer process influences defect distribution or anisotropy would help readers unfamiliar with additive manufacturing understand the study’s motivation.

Response 5: Thanks for your valuable suggestion. We have added the sentence in Section 1 “ Furthermore, specimen orientation still has an effect on the thermal diffusion and morphology of molten pool.”

Comments 6: In section 2.1, the experimental details mention cuboids and specimens, but their dimensions are not specified. 

Response 6: Thanks for your comments. We have specified the dimensions of cuboids and specimens in Section 2.1.

Comments 7: The tensile properties section (3.2) reports strength and ductility values, but it’s unclear how many tests were conducted per condition. 

Response 7: Thanks for your comments. We have specified the testing specimens for obtaining the tensile properties in Section 2.3 “Three specimens per orientation were tested for achieving the average strength/plasticity of the material.”

Comments 8: In the fatigue properties section (3.3), the method for determining fatigue limits (e.g., staircase method or curve fitting) is not specified. Clarifying this would aid in interpreting the reported values of 487.5 MPa and 562.5 MPa for AT specimens, and 687.5 MPa and 675 MPa for HIP specimens.

Response 8: Thanks for your comments. We have added the method for determining fatigue limits in Section 2.3 “The fatigue limit 107 cycles takes the mean value from unfractured specimens.”

Comments 9: The X-parameter model in section 3.4 is well-presented, but the manuscript does not explain how the fitting parameters α and β were determined. Additionally, discussing potential sources of error in the model (beyond the two aspects already noted) would provide a more balanced evaluation of its effectiveness.

Response 9: Thanks for your professional comments. We have explained the determination method of the fitting parameters α and β, and provided the program code as follows. Additionally, we also improve the influence factors for the X-parameter model.

import pandas as pd

import numpy as np

import matplotlib.pyplot as plt

from scipy.optimize import minimize

from sklearn.metrics import r2_score

data = pd.DataFrame({

    "Sigma": [575.0, 450.0, 514.7, 440.3, 588.2, 595.2, 572.7, 478.7, 498.4, 617.6, 588.2, 413.0, 542.1, 575.0, 550.0, 625.0, 576.7],

    "M": [41.3, 31.2, 65.3, 54.8, 71.0, 38.6, 50.2, 44.2, 60.4, 69.8, 42.7, 72.8, 40.4, 84.7, 134.5, 38.4, 142.4],

    "D": [1, 1, 1, 1, 1, 1, 1, 1, 1, 1, 1, 1, 1, 0.631, 1, 0.816, 0.836],

    "C": [0.468, 0.558, 0.626, 0.738, 0.518, 0.739, 0.490, 0.703, 0.938, 0.456, 0.726, 0.540, 0.715, 0.341, 0.167, 0.459, 0.211],

    "N": [377572, 354754, 91159, 284041, 49130, 55858, 123060, 143928, 103769, 41072, 90355, 312444, 199436, 6423121, 56607, 1846441, 1054956]

})

def formula_1(sigma, M, D, C, e, f):

    return sigma * (M ** (1/6)) * (D ** e) / (C ** f)

def objective(params):

    e, f = params

    X1 = formula_1(

        sigma=data["Sigma"].values,

        M=data["M"].values,

        D=data["D"].values,

        C=data["C"].values,

        e=e,

        f=f

    )

    lgN = np.log10(data["N"].values)

    a, b = np.polyfit(lgN, X1, 1)  

    X2 = a * lgN + b

    return -r2_score(X1, X2)

initial_params = [22.6, 2.29]

result = minimize(

    fun=objective,

    x0=initial_params,

    method="L-BFGS-B",

    options={"maxiter": 1000}

)

best_e, best_f = result.x

final_R2 = -result.fun

X1_opt = formula_1(

    sigma=data["Sigma"].values,

    M=data["M"].values,

    D=data["D"].values,

    C=data["C"].values,

    e=best_e,

    f=best_f

)

lgN = np.log10(data["N"].values)

a_opt, b_opt = np.polyfit(lgN, X1_opt, 1)

X2_opt = a_opt * lgN + b_opt  

plt.figure(figsize=(10, 6))

plt.scatter(lgN, X1_opt, color='blue', label='data point', alpha=0.7)  

plt.plot(lgN, X2_opt, color='red', linewidth=2, label=f'fitting curve (R²={final_R2:.4f})')  

plt.xlabel('log(N)')

plt.ylabel('X')

plt.title('fitting effect of X and log(N)')

plt.legend()

plt.grid(True, alpha=0.3)

plt.tight_layout()

plt.show()

output_df = data.copy()

output_df['log(N)'] = lgN.round(4)  

output_df['X(formula 1 calculation value)'] = X1_opt.round(2)  

output_cols = ['Sigma', 'M', 'D', 'C', 'N', 'log(N)', 'X(formula 1 calculation value)']

output_df = output_df[output_cols]

print("\n==================== detailed data output ====================")

print(output_df.to_string(index=False))

print(f"\n note:optimal parameter e={best_e:.4f}, f={best_f:.4f},maximum R²={final_R2:.4f}")

Comments 10: The references are relevant, but incorporating more recent studies on SLM Ti6Al4V fatigue behavior (e.g., from the last 2–3 years) could further contextualize your findings within the current state of the field.

Response 10: Thanks for your suggestion. Revised accordingly. We have added the more recent references.

Reviewer 3 Report

Comments and Suggestions for Authors

Revision for Mechanical properties and Fatigue life estimation of selective laser manufactured Ti6Al4V alloys in comparison of annealing treatment and hot isostatic pressing.

Gao et al. described the mechanical behavior and fatigue life of Ti6Al4V made by SLM, focusing on how annealing and HIP treatments affect properties. They examined how microstructure and defects affect strength, plasticity, and fatigue failure, considering the impact of build orientation. They conducted optical microscopy, SEM, XCT, static, and HCF testing. The main findings can be summarized as follows: strength drops as the α phase coarsens, but plasticity remains the same, defects such as micropores cause anisotropy and scatter in properties, which HIP helps reduce and fatigue life is correlated to defects, and the structure of α phases, with distinct trends for annealed and HIP samples. However, the manuscript has several areas for improvement, such as statistical analysis and a more detailed report of the testing. 

Minor
Fig 4c should have a white background for better readability.
What atmosphere gas was used? Besides Fig. 7b, Fig. 7e (lower left corner) shows a circular mark that seems like a trapped inert gas bubble.
Are printing parameters available? The beam speed may influence the solidification rate, hence grain size.
L105 should be high-cycle

Major
Did you use an extensometer for the static testing? Static curves are not included to illustrate the difference in strength and elasticity between printing directions and post-processing. It has been shown that HIP alters mechanical behavior, which can be observed macroscopically through the stress-strain curve and microscopically within the structure, with a change in phases evident through XRD. The static curves must be included so that the study is not fragmented into several microarticles.
Section 3.3 displays fatigue behavior. How many samples per stress level were tested? As in any experimental study, a statistical test is needed to quantify the data dispersion. Furthermore, the authors present an r^2 (correlation factor), but what is it correlated with? An adjustment to a type of S-Nf rule (the most known is Basquin) and coefficients must be presented and compared to literature (Pertuz 2023; Pagliari 2025) to respond to the problem raised in P2L60 (discrepancy and fatigue life estimation). 
The authors discussed microstructural features already described in the literature, such as the formation of acicular martensite during rapid cooling and the microstructural differences in different planes. However, as technologies evolve and new heat sources are developed, it is highly recommended to discuss differences and similarities with other PBF heat sources. While EBM may produce slower cooling and coarser grains due to the very rapid directionality of electron beam coils, yielding a higher fatigue performance linked to lower residual stress and coarser grain structure, SLM may lead to rapid cooling and finer grains may reduce fatigue performance due to higher residual stress. 

While the Murakami model appears to be an appropriate choice for defect quantification, I recommend that the authors consult Yin (Yin and Li, International Journal of Fatigue 2023) for a more detailed model that was tested using the same material and process.

***
Pagliari. High- and low-cycle fatigue behavior of additively manufactured Ti6Al4V and influence of surface finish. Engineering Failure Analysis. 2025, 109825. https://doi.org/10.1016/j.engfailanal.2025.109825

Pertuz. Strain-based Fatigue Experimental Study on Ti–6Al–4V Alloy Manufactured by Electron Beam Melting. Journal of Manufacturing and Materials Processing, 7 (1) 2023, https://doi.org/10.3390/jmmp7010025

English
The manuscript has some language issues mainly related to phrasing, clarity, and correctness of grammar and prepositions. A through check must be done.
"The approximate linear relationship of σmax-lgNf" for "The approximate linear relationship between σmax and lgNf."
"The type of micropore is caused by hollow powder or trapped gas within the molten pool" for "Micropores may be caused by hollow powder or trapped gas within the molten pool."

Author Response

Comments 1: Fig 4c should have a white background for better readability. What atmosphere gas was used? Besides Fig. 7b, Fig. 7e (lower left corner) shows a circular mark that seems like a trapped inert gas bubble. Are printing parameters available? The beam speed may influence the solidification rate, hence grain size. L105 should be high-cycle.

Response 1: Thanks for your suggestion. Modified accordingly. In Fig. 4c, it reveals the extracted three-dimentional morphology of defects in a given volume. In fact, the alter of X-ray imaging background cannot bring significant improvement for such minor defects, and more importantly, it should be pay attention to magnified morphology of defects. An inert Argon gas protection atmosphere was used in Section 2.1. In Fig. 7e, a circular micropore can be found, however, it can be neglected in comparison to such a large LOF served as crack initiation site. Process parameters was verified in advance for achieving specimens with approximately fully density.

Comments 2: Did you use an extensometer for the static testing? Static curves are not included to illustrate the difference in strength and elasticity between printing directions and post-processing. It has been shown that HIP alters mechanical behavior, which can be observed macroscopically through the stress-strain curve and microscopically within the structure, with a change in phases evident through XRD. The static curves must be included so that the study is not fragmented into several microarticles.

Response 2: Thanks for your suggestion. An extensometer was used for testing gauge strain of specimens in Section 2.3. The stress-strain curve was added in Fig. 5a, and the correspondence text was modified.

Comments 3: Section 3.3 displays fatigue behavior. How many samples per stress level were tested? As in any experimental study, a statistical test is needed to quantify the data dispersion. Furthermore, the authors present an r^2 (correlation factor), but what is it correlated with? An adjustment to a type of S-Nf rule (the most known is Basquin) and coefficients must be presented and compared to literature (Pertuz 2023; Pagliari 2025) to respond to the problem raised in P2L60 (discrepancy and fatigue life estimation). 

Response 3: Thanks for your valuable suggestion. In this study, two samples of of the SLM annealed and HIPed specimens per orientation at each stress level were basically tested. We have raised the fatigue life data of limited samples at the stress ratio R of 0.1 in the AT and HIP status. However, the testing data has a poor linear fitting due to few sample numbers in the AT status, thus we have deleted fitting curve. The linear fitting formula and its correlation factor r^2 related to the specimens in the HIP status was provided in Fig. 6.

Comments 4: The authors discussed microstructural features already described in the literature, such as the formation of acicular martensite during rapid cooling and the microstructural differences in different planes. However, as technologies evolve and new heat sources are developed, it is highly recommended to discuss differences and similarities with other PBF heat sources. While EBM may produce slower cooling and coarser grains due to the very rapid directionality of electron beam coils, yielding a higher fatigue performance linked to lower residual stress and coarser grain structure, SLM may lead to rapid cooling and finer grains may reduce fatigue performance due to higher residual stress. 

Response 4: Thanks for your valuable suggestion. In this study, the mechanical properties of SLM manufactured specimens in the AT and HIP was the key point rather than the comparison of PBF heat sources. Experimental studies [1,2] have previously reported that the comparison of the microstructures and mechanical properties of Ti-6Al-4V fabricated by SLM and EBM.

[1] Zhao X.L., Li S.J., Zhang M., et al. Comparison of the microstructures and mechanical properties of Ti-6Al-4V fabricated by selective laser melting and electron beam melting. Materials & Design. 2016, 95: 21-31.

[2] Günther J., Krewerth D., Lippmann T., et al. Fatigue life of additively manufactured Ti-6Al-4V in the very high cycle fatigue regime. International Journal of Fatigue, 2016, 94: 236-245.

Comments 5: While the Murakami model appears to be an appropriate choice for defect quantification, I recommend that the authors consult Yin (Yin and Li, International Journal of Fatigue 2023) for a more detailed model that was tested using the same material and process.

Pagliari. High- and low-cycle fatigue behavior of additively manufactured Ti6Al4V and influence of surface finish. Engineering Failure Analysis. 2025, 109825. https://doi.org/10.1016/j.engfailanal.2025.109825

Pertuz. Strain-based Fatigue Experimental Study on Ti–6Al–4V Alloy Manufactured by Electron Beam Melting. Journal of Manufacturing and Materials Processing, 7 (1) 2023, https://doi.org/10.3390/jmmp7010025

Response 5: Thanks for your valuable suggestion. We have referred to the recommended literature. Indeed, the Murakami model was an appropriate choice for defect quantification, but still not suitable for fatigue life prediction depending on the α colony in the HIP status.. Future work will address this important topic.

Comments 6: The manuscript has some language issues mainly related to phrasing, clarity, and correctness of grammar and prepositions. A through check must be done.
"The approximate linear relationship of σmax-lgNf" for "The approximate linear relationship between σmax and lgNf."
"The type of micropore is caused by hollow powder or trapped gas within the molten pool" for "Micropores may be caused by hollow powder or trapped gas within the molten pool."

Response 6: We are very sorry for our poor English writing. We have pursued a native English scholar carefully checked the paper, correct grammar and rephrase with text that make it more accurate and concise. The modified version has been significantly improved from the perspective of logicality and readability.

Round 2

Reviewer 2 Report

Comments and Suggestions for Authors

The authors have addressed most of the initial comments effectively by adding quantitative data, clarifying methods, improving readability, and updating references.

However, some responses could be enhanced with minor clarifications or expansions, as noted above, to improve transparency and reader comprehension. The deletion of the fatigue curve in Fig. 6 is acceptable given the poor fit, but it highlights a need for cautious interpretation of the fatigue data until further testing can be conducted.

I recommend accepting the manuscript with minor revisions. The current version is significantly improved and suitable for publication once the following minor adjustments are made:

  • Specify the measurement method for microstructure data and whether values are statistically representative.
  • Briefly explain why the X-parameter model is inapplicable to HIP (e.g., defect vs. microstructure initiation).

  • Expand the build orientation sentence to link thermal diffusion and molten pool morphology to defect distribution or anisotropy.

  • Add a short explanation of the X-parameter optimization process and parameter significance.

Author Response

Comments 1: Specify the measurement method for microstructure data and whether values are statistically representative.

Response 1: Thanks for your valuable comments. The metallographic preparation method is a common approach for Ti6Al4V alloy, and more metallography were observed carefully, not limited to a single specimen in the AT and HIP status. In this study, the representative metallographic images are presented for analyzing microscopic characteristics, and the measured values are representative.

Comments 2: Briefly explain why the X-parameter model is inapplicable to HIP (e.g., defect vs. microstructure initiation).

Response 2: Thanks for your professional comments. The authors did indeed ignore the issue and did not explain this clearly. As the effect of HIP, the fact that the alteration of fatigue crack initiation sites from defects to α colony was clarified from the fatigue fracture images observation. Due to the existence of these defects, the fatigue initiation life comprised of the formation and propagation of small crack can be neglected in Section 3.3 (Page 9, line 237), which is suitable for the X-parameter model in terms of the fatigue stable propagation life. However, α colony served as fatigue crack initiation site makes the above situation more complicated, since the total fatigue life consists of the crack initiation and propagation stages in this case (Page 9, line 243), resulting in an approximate linear relationship between σmax and lgNf. This relationship between α colony characteristics and fatigue life needs further study. We have modified the headline in Section 3.4 to “Defect-based fatigue life evaluation”.

Comments 3: Expand the build orientation sentence to link thermal diffusion and molten pool morphology to defect distribution or anisotropy.

Response 3: Thanks for your comments. Modified accordingly. “Furthermore, specimen orientation still has an effect on the thermal diffusion and morphology of molten pool due to different deflection angles of laser beam, which often induces more defects in a larger melting layer.”

Comments 4: Add a short explanation of the X-parameter optimization process and parameter significance.

Response 4: Thanks for your comments. The X-parameter optimization process and parameter significance were introduced in Section 3.4.

Reviewer 3 Report

Comments and Suggestions for Authors

A revised version was submitted.

Some minor revisions are needed

Please include the extensometer model within the text

Author Response

Comments 1: Please include the extensometer model within the text.

Response 1: Thanks for your comments. We have added the extensometer model.